# Metastatic prostate cancer men's attitudes towards treatment of the local tumour and metastasis evaluative research (IP5-MATTER): protocol for a prospective, multicentre discrete choice experiment study

Martin John Connor [1,2] Mesfin G Genie [3,4] Michael Gonzalez,[5] Naveed Sarwar,[5] Kamalram Thippu Jayaprakash [6,7] Gail Horan,[6] Feargus Hosking-Jervis,[1] Natalia Klimowska-Nassar,[1,8] Johanna Sukumar,[1,8] Tzveta Pokrovska,[5] Dolan Basak,[5] Angus Robinson,[9] Mark Beresford,[10] Bhavan Rai,[11] Stephen Mangar,[5] Vincent Khoo,[12] Tim Dudderidge,[13] Alison Falconer,[5] Mathias Winkler,[1,2] Verity Watson,[3] Hashim Uddin Ahmed [1,2]

VW and HUA are joint senior authors.

For numbered affiliations see end of article.

**Correspondence to**
Dr Martin John Connor;
m.connor@imperial.ac.uk

## ABSTRACT

**Introduction** Systemic therapy with androgen deprivation therapy (ADT) and intensification with agents such as docetaxel, abiraterone acetate and enzalutamide has resulted in improved overall survival in men with *de novo* synchronous metastatic hormone-sensitive prostate cancer (mHSPC). Novel local cytoreductive treatments and metastasis-directed therapy are now being evaluated. Such interventions may provide added survival benefit or delay the requirement for further systemic agents and associated toxicity but can confer additional harm. Understanding men's preferences for treatment options in this disease state is crucial for patients, clinicians, carers and future healthcare service providers.

**Methods** Using a prospective, multicentre discrete choice experiment (DCE), we aim to determine the attributes associated with treatment that are most important to men with mHSPC. Furthermore, we plan to determine men's preferences for, and trade-offs between, the attributes (survival and side effects) of different treatment options including systemic therapy, local cytoreductive approaches (external beam radiotherapy, cytoreductive radical prostatectomy or minimally invasive ablative therapy) and metastases-directed therapies (metastasectomy or stereotactic ablative body radiotherapy). All men with newly diagnosed mHSPC within 4 months of commencing ADT and WHO performance status 0–2 are eligible. Men who have previously consented to a cytoreductive treatment or have developed castrate-resistant disease will be excluded. This study includes a qualitative analysis component, with patients (n=15) and healthcare professionals (n=5), to identify and define the key attributes associated with treatment options that would warrant trade-off evaluation in a DCE. The main phase component planned recruitment is 300 patients over 1 year, commencing in January 2021, with planned study completion in March 2022.

### Strengths and limitations of this study

► IP5-MATTER will be the first multicentre discrete choice experiment conducted to evaluate men's treatment preferences for local cytoreductive and metastasis-directed therapy in metastatic hormone-sensitive prostate cancer (mHSPC).

► The study will explore preference heterogeneity according to the patients' personal characteristics.

► This is the first study to determine if men with mHSPC are willing to accept the potential effect sizes that are reported in randomised studies.

► It is not feasible to incorporate all factors that may affect patients' treatment preferences (eg, impact of hospital visits; the impact of drug effectiveness on disease progression).

► This study will be undertaken in the UK and may not be generalisable to other men with mHSPC in other countries.

**Ethics and dissemination** Ethical approval was obtained from the Health Research Authority East of England, Cambridgeshire and Hertfordshire Research Ethics Committee (Reference: 20/EE/0194). Project information will be reported on the publicly available Imperial College London website and the Heath Economics Research Unit (HERU website including the HERU Blog). We will use the social media accounts of IP5-MATTER, Imperial Prostate London, HERU and the individual researchers to disseminate key findings following publication. Findings from the study will be presented at national/international conferences and peer-reviewed journals. Authorship policy will follow the recommendations of the International Committee of Medical Journal Editors.

**Trial registration number** NCT04590976.

## INTRODUCTION

The number of men diagnosed with prostate cancer each year in the UK is reported to be 47 000.[1] In this cohort, it is estimated that 4500 men will have a diagnosis of *de novo* synchronous metastatic hormone-sensitive prostate cancer (mHSPC) at index presentation.[1] The mainstay of treatment for such men has been hormonal castration with androgen deprivation therapy (ADT) alone. In isolation, this treatment modality results in the emergence of a castrate-resistant state within a median of 11–18 months that restricts overall survival (OS) to 3.5 years.[2 3]

Recent advances with using ADT in combination with docetaxel or novel anti-androgens (eg, enzalutamide, abiraterone acetate, apalutamide) at diagnosis have led to a rise in the median OS to 4.8 years.[3–9] Current research is considering the role of local cytoreductive approaches (cytoreductive radical prostatectomy, external beam radiotherapy or minimally invasive ablative therapy) and metastasis-directed interventions (predominantly, surgical metastatectomy or stereotactic ablative body radiotherapy (SABR)) with a view to acquiring further oncological benefit in this setting.[1 10–18]

Research into the role of local cytoreductive and metastasis-directed interventions has, in part, been propagated by the emergence of the principle of an 'oligometastatic' state.[19] In this context, men are hypothesised to harbour a favourable clinically defined, low disease burden that lies between 'locally advanced' and 'polymetastatic' disease.[19–21] Early randomised data suggest that men who exhibit such disease may gain the most benefit from additional cytoreductive interventions.[10 22]

These interventions offer potential overall and progression-free survival benefits. Furthermore, they may allow second-line systemic agents and their associated toxicities to be avoided or deferred. However, we currently understand very little about men's treatment preferences and the decision-making process in this emerging treatment paradigm.[23]

The IP5-MATTER study will use a discrete choice experiment (DCE) to determine men's preferences for, and trade-offs between, the attributes (survival and side-effects) of different treatment options for mHSPC including systemic therapy, local cytoreductive and metastases-directed therapies.

### Discrete choice experiment

A DCE is a survey method that is related to 'conjoint analysis'.[24 25] A DCE was selected as the method for this study as it allows the researcher to calculate the trade-offs that patients make between specific treatment options.[26] The trade-off in IP5-MATTER is whether a man is willing to accept the disadvantages to himself (ie, side effects, risk, visits to hospitals) with an extra treatment(s) in exchange for potential survival or cancer progression-free survival benefits. This is also termed the 'marginal rate of substitution'.[26]

In the IP5-MATTER study, patients will be presented with hypothetical treatment scenarios.[26] Each scenario will be described using a set of 'attributes' associated with each treatment for example, treatment modality, urine incontinence, fatigue, loss of erections, length of time until cancer starts to grow again, length of survival after diagnosis. Each treatment scenario will be associated with a realisation or level for each attribute. For instance, in a hypothetical treatment the urine incontinence attribute may take the value of 10% or 1 in 10 men, that is, of 100 men who have the hypothetical treatment 1 in 10 will experience urine incontinence.

The attributes and levels will be combined into hypothetical treatment scenarios and these will be grouped into treatment choice sets using experimental design theory. In essence, the patient will be asked to 'choose' between two treatments A or B in a choice set.

Each patient will be asked to complete several choice sets each containing different pairs of treatments. Each choice set will vary the levels of the attributes in a systematic way that allows the researcher to identify how the different realisations of the attributes affects the patients' choices (figure 1). We will include up to 12 choices per patient.[24 25]

### Previous DCEs in prostate cancer

A number of DCE studies have previously been completed in the field of non-metastatic prostate cancer, metastatic castrate-resistant prostate cancer (mCRPC) and mHSPC suitable to elicit patients' treatment preferences.[27–31]

#### Non-metastatic prostate cancer

Sculpher *et al* undertook the first prostate cancer DCE in 129 men in the UK with non-metastatic prostate cancer (T1 or T2).[31] The primary focus was to evaluate side effects of various ADT options. The group reported men were willing to forgo up to 3 months of life expectancy to avoid limitations in physical energy.[31]

The recent COMPARE (COMparing treatment options for ProstAte cancer) multicentre study focused on 650 men with newly diagnosed localised non-metastatic prostate cancer (NCT01177865).[32] Watson and colleagues identified that patients' preferences for attributes for treatment were, in order: survival, no incontinence, not needing further treatment and maintaining an erection. Interestingly, men were willing to trade 0.68% absolute survival for a 1% absolute improvement in urinary function and 0.28% absolute survival for 1% absolute improvement in maintaining erections.[32]

Furthermore, King and colleagues in a study of 357 men with localised prostate cancer, who had undergone either radical therapy or commenced ADT, found men were similarly willing to trade-off 6 months of survival if they could avoid erectile dysfunction and approximately 13 months of survival if they could avoid urinary incontinence.[28]

Finally, in a study of 110 men with early prostate cancer, as well as their treating urologist, conducted by de Bekker-Grob and colleagues it was reported that while urinary incontinence was a concern among both patient

| TREATMENT FEATURES | Treatment A | Treatment B |
|---|---|---|
| How your metastatic prostate cancer is managed | Hormonal therapies (with or without chemotherapy) followed by **no additional treatment** to prostate | Hormonal therapies (with or without chemotherapy) followed by **an overnight-stay surgery** and **4 weeks** of recovery time |
| Is specialised radiotherapy to cancer deposits included? | No | Yes, there are hospital appointments every day for one week to receive this specialist radiotherapy. |
| How long a man, on average, is expected to live after the diagnosis? | 50 months | 70 months |
| How long, on average, until the cancer starts to grow again? | 40 months | 60 months |
| Proportion of men who have permanent urinary incontinence after treatment (%) | 5% (5 out of 100) | 20% (20 out of 100) |
| The proportion of men who are <u>not</u> able to maintain an erection sufficient for intercourse (%) | 40% (40 out of 100) | 70% (70 out of 100) |
| The proportion of men who have a feeling of extreme tiredness (fatigue) impacting on daily activities (%) | 10% (10 out of 100) | 40% (40 out of 100) |
| **Which treatment option would you choose?** | ☐ | ☐ |

**Figure 1** Example choice task. Example choice task for this study that is subject to change following the completion of Phase I and Phase II.

and clinician, the risk of erectile dysfunction secondary to radiotherapy was solely valued by the clinician alone.[27]

### Metastatic castrate-resistant prostate cancer

In the mCRPC setting, Eliasson and colleagues study 285 men with mCRPC from three countries (Germany, France and the UK).[30] The group reported that lower fatigue (OR, 1.365 (95% CI: 1.219 to 1.528)), and fewer additional hospital visits (OR, 1.245 (95% CI: 1.111 to 1.397)) were the most important factors affecting any treatment choice.[30]

Furthermore, Uemura and colleagues studied 133 Japanese men with mCRPC to demonstrate that fatigue (relative importance (RI)=24.9% (95% CI: 24.7 to 25.1)) was the most important attribute.[33] This was followed by a reduction in the risk of bone pain (RI=23.2% (95% CI: 23.0 to 23.5)). The authors concluded that men were more concerned by quality-of-life secondary to side effects than any potential extension in survival.[33]

### Metastatic hormone-sensitive prostate cancer

A single DCE in mHSPC has been performed by de Freitas and colleagues on a mixed cohort of 152 men, of which a minority (45/152; 29.6%) had mHSPC. The group evaluated docetaxel or abiraterone acetate in combination with ADT. Within the mHSPC subgroup, the primary ranked attribute was drug effectiveness on survival (relative attribute importance (RAI)=9.08), followed by pain control (RAI=6.37). Of note, men in this cohort were least concerned about method of treatment delivery (RAI=1.51).[29]

In summary, to our knowledge, no studies have undertaken a DCE to elicit men's preferences for systemic therapy in combination with either local cytoreductive and/or metastasis-directed treatments. The aforementioned, DCE studies in prostate cancer (non-metastatic, metastatic castrate-resistant and mHSPC) have predominantly highlighted patients' preferences in the context of being offered established systemic therapy options and a select number of bone-targeted agents.

The IP5-MATTER study is a multicentre DCE designed to evaluate novel treatments (cytoreductive prostatectomy, external beam radiotherapy, minimally invasive ablative therapy and SABR) in addition to systemic therapy for the first time. To appropriately invest in emerging cytoreductive prostate and metastasis-directed treatments options that are most acceptable to such patients, we believe that an attempt to formalise our understanding of the trade-offs between oncological benefits and risks in this cohort of patients should be performed.

### Study hypothesis

The null hypothesis ($H_0$) is that men with *de novo* synchronous hormone-sensitive metastatic prostate cancer will accept the complications and side effects associated with combined systemic and cytoreductive local prostate and/or metastasis-directed therapy for any potential oncological benefit.

The scientific hypothesis ($H_1$) is that men with *de novo* synchronous hormone-sensitive metastatic prostate cancer may not accept the complications and risk of side effects associated with cytoreductive local prostate and/or metastasis-directed therapy for any potential oncological benefit.

### Patient and public involvement

A patient involvement interview was held with two patients who had metastatic prostate cancer to determine initial patient acceptability of the study. The integrated qualitative component of this study (Phases 1 and 2) incorporates patient involvement. A patient and public involvement representative was present during the HRA REC assessment. This representative will continue to be involved throughout the duration of the study. This will include the Trial Management Group. Furthermore, other patients that are not involved directly in this study will be present on the independent Trials Steering Committee.

## METHODS AND ANALYSIS
### Study design

IP5-MATTER will be performed at sites within the UK. This will be a cross-sectional questionnaire study. The initial qualitative phase (Phase 1) will involve semistructured interviews with (up to 5) clinician members of the multidisciplinary or tumour board meeting and (up to 5) patients who have had a diagnosis of mHSPC, to identify and define the key attributes associated with treatment options that would warrant trade-off evaluation. These will be used to create the first version of the questionnaire (Phase 1).[26]

Interviews with participants will be audiorecorded and transcribed verbatim. These will be destroyed on transcription or within 2 months (whichever occurs earliest). Transcripts will be analysed using an inductive thematic analysis by at least two researchers. Inductive thematic analysis will be performed as described by Nowell and colleagues six-step iterative process.[34] Researchers will (1) familiarise themselves with the data (eg, document potential themes/codes), (2) generate initial codes (eg, reflective journaling), (3) search for themes (eg, triangulation of data), (4) review themes (eg, test for referential adequacy to raw data), (5) define themes and name and (6) produce a final report.[34] The responses and suggestions from the interviews will be used to develop the questionnaire in order to ensure that (1) any attribute or levels that do not seem appropriate are removed or modified; (2) any additional attributes or levels that are not in the questionnaire are included; (3) the questionnaire is understandable and (4) any complex treatment information is explained in a accessible manner.[26]

In the DCE questionnaire, the attributes will be combined into different hypothetical scenarios that will be presented to men in the form of a treatment choice. The hypothetical scenarios included in the questionnaire

will be selected using experimental design theory. These attributes and levels will be combined to create treatment scenarios and paired into choice sets of two scenarios. We will use NGENE software (ChoiceMetrics) to generate a 36 choice tasks D-efficient design with non-informative (null) priors and allowing estimation of non-linear effects of attributes.[35] The design is based on the main effects only (ie, without interactions). The 36 choice tasks will be split into three blocks of 12 tasks to reduce respondent burden. Respondents will be allocated to one of the three blocks. Risk attributes will be presented using icon arrays, ratios and percentages to ease comprehension (figure 1) shows an example choice task. The questionnaire will be trialled on a different group of up to 10 men using a semistructured 'Think Aloud' interview and any further changes made to create the final version of the DCE questionnaire (Phase 2).[36] The final survey instrument will also collect information on patients' characteristics (eg, age, education).

This final questionnaire will then be used in the main phase of the study in 300 men across a number of UK trial sites over 12 months (Phase 3). The DCE questionnaire will be given to the participants to fill in, either directly following a face-to-face hospital clinic appointment or in their own home. For the electronic version of the survey, REDCap software will be used, with tablets available following a face-to-face hospital clinic appointment where required.

## Study population

Men diagnosed with *de novo* synchronous mHSPC who have not consented to a form of local cytoreductive or metastasis-directed therapy.

## Eligibility

### Inclusion criteria

▶ Diagnosed with mHSPC within 4 months of screening visit.
▶ WHO performance status 0–2.

### Exclusion criteria

▶ Castrate-resistant metastatic prostate cancer.
▶ Patient has consented to a form of local cytoreductive treatment to the prostate.
▶ Patient has consented to a form of metastasis-directed therapy.

## Identification of patients and consent

All men diagnosed with prostate cancer discussed at the multidisciplinary team (MDT) meeting (or a tumour board) as well as any man meeting the eligibility criteria prior to the MDT meeting will be identified for screening. Members of the MDT will identify patients suitable for IP5-MATTER. The treating clinicians will mention the study and the local research nurses, research fellows, clinical trial coordinators, clinical trial practitioners or the treating clinicians will then approach the patient if they are interested.

Patients with confirmed metastatic prostate cancer and who satisfy the entry criteria will be approached for study enrolment. Patients can be included even if they have already commenced ADT, chemotherapy or an alternative hormonal agent as part of their standard of care. They will be provided with a patient information sheet (PIS) and as much time as they might need to consider whether or not they wish to participate (minimum 24 hours).

Patients already aware of their diagnosis can be approached by telephone to enquire as to their interest in the study so that a PIS can be sent out by email or post prior to a clinical visit.

In the qualitative phases (Phases 1 and 2), patients who wish to participate after reading the PIS will be given a date for the semistructure interview (face-to-face or using audio-visual software). In the main Phase 3 component, patients who meet the eligibility criteria and are willing to participate will be given the questionnaire to fill in their own time either in the face-to-face or virtual hospital clinic or in their own home. Remote informed consent can be undertaken with the assistance of a telephone or audio-visual software, determined by local availability and protocol. An electronic method (in-built REDCap e-consent form mirroring paper version with 'advanced electronic signature' using a smartphone, tablet, computer) will enable remote informed consent in accordance with HRA-MHRA e-consent guidance (2018).[37] Informed consent will be signed before implementation of any study-related procedure.

## Randomisation

No randomisation is required for this study.

## Study outcome measures

### Primary outcomes

▶ To determine the attributes associated with treatment that are most important to men with mHSPC.
▶ To determine men's preferences for, and trade-offs between, the attributes (survival and side effects) of different treatment options in metastatic prostate cancer including systemic therapy, local cytoreductive and metastases-directed therapies.

### Secondary outcomes

▶ To inform treatment pathways that are being developed nationally and internationally following randomised trials that are being conducted to evaluate these questions.
▶ The effect size from these studies can be used with our IP5-MATTER study to determine if, on average, men are willing to accept the potential effect sizes that are demonstrated in trials.

### Study visits

▶ Phase 1 or 2: single-visit for semistructured interview or review of electronic study-developed questionnaire (table 1).

**Table 1** Visit schedule

| IP5-MATTER | | |
|---|---|---|
| **Visit** | **Screening** | **Visit 1** |
| **Week** | **0** | **0** |
| Informed consent | X | |
| Inclusion and exclusion | X | |
| Demographics | X | |
| Medical history | X | |
| Study intervention: discrete choice experiment | | X |

This study does not have a follow-up visit.

▶ Phase 3 (main study): single-visit completion of the electronic DCE during a routine hospital visit or at patient's own home (table 1).

### Follow-up
There is no planned follow-up for this study.

### Sample size calculation and statistical analyses
#### Sample size and power considerations
IP5-MATTER trial sample size is set at 300. This is based on the premise that the precision of DCE parameter estimates improves until a sample size of 300 and then further gains in precision are small, according to the sampling strategy recommendations set by the International Society for Pharmacoeconomics and Outcomes Research Good Research Practices for Conjoint Analysis Task Force (2011).[24]

#### Planned recruitment rate
We are aware from the IP2-ATLANTA trial that centres have approximately three men diagnosed with metastatic prostate cancer per month.[12 38] The initial qualitative phase will focus on the Northwest London regional cancer network (three hospitals) and take 6 weeks to complete. Full phase recruitment is based on 300 patients across 30 sites recruiting a minimum of 10 patients per year or about one patient per month. This is an estimated recruitment rate of 26%. Our previous DCEs (ie, COMPARE) had an acceptance rate of over 50%.[32]

#### Statistical analysis
Responses from the DCE will be analysed using multinomial logit (MNL) regression techniques that allow for multiple observations from each respondent and preference differences across respondents.[26 32] This will be used to understand treatment preferences and trade-offs made by patients when considering treatment options of mHSPC. In the MNL analysis, the functional form is specified as:

$$U_{ntj} = \beta X_{ntj} + \varepsilon_{ntj}$$

which represents the utility of option $j$ in choice task $t$ for patient $n$, where $X_{ntj}$ is a vector of variables representing the levels of the treatment presented in option $j$, $\beta$ is a vector of utility weights associated with each attribute and $\varepsilon_{ntj}$ is the error term. In the analysis, we will assume that patients gain utility (welfare) from treatment and the amount of welfare depends on the attributes and levels of the treatment. We assume that in each choice set patients choose the treatment that would bring them the highest welfare.

From the results of the DCE, it is possible to estimate the RI of attributes and the trade-offs that individuals are willing to make between attributes of a treatment. These trade-offs are how much of one attribute respondents are willing to sacrifice for improvements in another.[26]

In this case, we can explore how much life expectancy patients must gain before they are willing to accept treatment side effects. Further, based on the regression results, we can calculate whether the attribute had a positive or negative effect on respondent utility, and the RI of a unit change in each attribute when respondents are choosing between treatments. We can also calculate the trade-offs that respondents make between treatment attributes when making choices, which is represented by the ratio of the coefficients.[32]

Additionally, we will explore observed preference heterogeneity according to patients' characteristics. The observed characteristics will be analysed by interacting them with the attributes in the MNL model specifications.

### Data collection
The principal means of data collection from participants will be electronic data capture using the web-based REDCap generated platform, provided by the Sponsor. All eCRFs will be completed using de-identified data.

### Data monitoring and archiving
On two occasion per year, a combined independent data monitoring and trial steering committee will meet. All trial documentation will be archived for a minimum of 10 years following the completion of the study.

### Trial funding, organisation and administration
IP5-MATTER was approved by Health Research Authority East of England, Cambridgeshire and Hertfordshire Research Ethics Committee (20/EE/0194). The study is a joint collaboration between Imperial College London and Health Economic Research Unit (HERU), University of Aberdeen. This study is funded by the Wellcome Trust (204998/Z/16/Z) and University College London Hospitals (UCLH) Charity (P83624/1348).

The study will be monitored periodically by trial monitors to assess progress and verify adherence to the protocol, ICH GCP E6 guidelines and other national/international requirements and to review the completeness, accuracy and consistency of the data (online supplemental material). The study may be subject to inspection and audit by regulatory bodies to ensure adherence to GCP and the UK Policy Framework for Health and Social Care Research.

## DISCUSSION

IP5-MATTER is a multicentre, prospective DCE in men with a new diagnosis of de novo synchronous mHSPC. This is the first DCE, to the author's knowledge, that evaluates men's treatment preferences for local cytoreductive and metastasis-directed therapy following such a diagnosis.

This study will determine men's preferences for, and trade-offs between, the attributes (survival and side effects) of different treatment options in metastatic prostate cancer including systemic therapy, local cytoreductive and metastases-directed therapies.

When taken collectively, it is expected that the results will be used to determine if, on average, men are willing to accept the potential effect sizes that are reported in current and future randomised studies. This will have wide-reaching implications on the investment in, and provision of, specific treatment options proposed for integration into the current standard of care treatment pathways for men with mHSPC.

Furthermore, it is possible that once such treatment pathways are established the findings from this DCE may be integrated into further work towards the creation of a decision treatment aid (DAT) for men with mHSPC. There is residual uncertainty in the role of DCE's in the development of such DATs.[39] However, this methodology is currently being explored in parallel to this study in localised prostate cancer and other benign surgical settings, and if proven IP5-MATTER may offer utility in the development in any future DAT for this cohort.[40 41]

## CONCLUSION

IP5-MATTER answers important questions relating to men's decision-making in relation to local cytoreductive and metastasis-directed treatments following a new diagnosis of mHSPC.

## TRIAL STATUS

IP5-MATTER is opened to recruitment in 30 centres in England, Wales and Scotland in February 2022 and planned recruitment completed by February 2022 and study completion in April 2022.

**Author affiliations**
[1]Imperial Prostate, Divison of Surgery, Department of Surgery and Cancer, Imperial College London, London, UK
[2]Imperial Urology, Charing Cross Hospital, Imperial College Healthcare NHS Trust, London, UK
[3]Health Economics Research Unit (HERU), Faculty of Medicine, University of Aberdeen, Aberdeen, UK
[4]Economics, Ca' Foscari University of Venice, Venezia, Italy
[5]Department of Oncology, Charing Cross Hospital, Imperial College Healthcare NHS Trust, London, UK
[6]Department of Oncology, Addenbrooke's Hospital, Cambridge, UK
[7]Department of Oncology, Queen Elizabeth Hospital King's Lynn NHS Foundation Trust, King's Lynn, UK
[8]Imperial College Clinical Trials Unit (ICTU), Imperial College London, London, UK
[9]Department of Oncology, Brighton and Sussex University Hospitals NHS Trust, Brighton, UK
[10]Department of Oncology, Royal United Hospitals Bath NHS Foundation Trust, Bath, UK
[11]Department of Urology, Newcastle Upon Tyne Hospitals NHS Foundation Trust, Newcastle Upon Tyne, UK
[12]Department of Oncology, The Royal Marsden NHS Foundation Trust and Institute of Cancer Research, London, UK
[13]Department of Urology, University Hospital Southampton NHS Foundation Trust, Southampton, UK

**Acknowledgements** We would like to thank all the participants, study PIs, trial clinicians, research nurses, Imperial Clinical Trial Unit staff and other site staff who have been responsible for setting up, recruiting participants and collecting the data for the IP5-MATTER trial. We are also grateful for the ongoing support of the Trial Management Group and our IP5-MATTER patient representatives. Finally, we would like to thank our trial funder the Wellcome Trust and University College London Hospitals (UCLH) Charity.

**Contributors** MJC, MG, FH-J, MW, VW, HUA contributed to conception and design of IP5-MATTER trial. All authors have read and approved the final manuscript.

**Funding** MJC's research is support by University College London Hospitals (UCLH) Charity and the Wellcome Trust. Mesfin Genie and Verity Watson are based at the Health Economics Research Unit (HERU), University of Aberdeen. HERU is funded by the Chief Scientists Office of the Scottish Government Health and Social Care Directorate. KTJ acknowledges research grant from the UK National Institute of Health Research Clinical Research Network Eastern and has received educational grants from Bayer UK, Janssen Oncology, Pfizer, Roche, and Takeda. HUA's research is supported by core funding from the United Kingdom's National Institute of Health Research (NIHR) Imperial Biomedical Research Centre.

**Competing interests** HUA currently receives funding from the Wellcome Trust, Prostate Cancer UK, MRC (UK), Cancer Research UK, The Urology Foundation, BMA Foundation, Imperial Healthcare Charity, Sonacare, Trod Medical and Sophiris Biocorp for trials in prostate cancer. HUA was a paid medical consultant for Sophiris Biocorp, Sonacare and BTG in the past 3 years. He is currently a paid proctor for Sonacare and Boston.

**Patient consent for publication** Not applicable.

**Provenance and peer review** Not commissioned; externally peer reviewed.

**ORCID iDs**
Martin John Connor http://orcid.org/0000-0003-4033-7508
Mesfin G Genie http://orcid.org/0000-0002-1744-4666
Kamalram Thippu Jayaprakash http://orcid.org/0000-0001-7217-4593
Hashim Uddin Ahmed http://orcid.org/0000-0003-1674-6723

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
