## [Reviewer comments · BMJ Open]

ARTICLE DETAILS

TITLE (PROVISIONAL)	Metastatic prostate cancer men's Attitudes towards Treatment of the local Tumour and metastasis Evaluative Research (IP5-MATTER): Protocol for a prospective, multicentre discrete choice experiment study
AUTHORS	Connor, Martin; Genie, Mesfin G; Gonzalez, Michael; Sarwar, Naveed; Thippu Jayaprakash, Kamalram; Horan, Gail; Hosking-Jervis, Feargus; Klimowska-Nassar, Natalia; Sukumar, Johanna; Pokrovska, Tzveta; Basak, Dolan; Robinson, Angus; Beresford, Mark; Rai, Bhavan; Mangar, Stephen; Khoo, Vincent; Dudderidge, Tim; Falconer, Alison; Winkler, Mathias; Watson, Verity; Ahmed, Hashim

VERSION 1 – REVIEW

REVIEWER	Lane, Giulia University of Michigan
REVIEW RETURNED	12-Apr-2021

GENERAL COMMENTS	The authors describe the protocol for a discrete choice experiment (DCE) to determine preferences and values for treatment decisions among men with metastatic, hormone sensitive prostate cancer. The study is clearly described, it is innovative and will move the field of shared decision making forward. Some minor suggestions for the authors to consider are listed below: 1) Page 9 Line 264-283. Qualitative interviews will be used to determine the attributes for the DCE during Phase 1 & 2. It would be helpful to readers to add more detail about the DCE Questionnaire properties (Phase 3). Will there be three levels of options per attribute? How will these levels be determined? Will a specific software be used to create and administer the DCE Questionnaire? 2) Page 11 Line 325. Men will be given the DCE questionnaire to fill in their own time either in the face-to-face or virtual hospital clinic or in their own home. How will patients access to the DCE during clinic encounters or in their own home -- ie will this be a web-based questionnaire? How will the study ensure equal access to patients that do not have access to the internet (will computer access be provided during in person clinic appointments)? 3) Page 16, Line 474. The trial status is described as follows "IP5-MATTER is planning to open...in January 2021." This text may benefit from updating prior to publication to reflect the current study status - (i.e. whether the trial opened in January 2021 as expected or whether the start date has been delayed).
---

	4) Page 15, Line 461-465. The results of this study certainly have implications for healthcare providers, clinicians and patients. The authors can consider expanding the discussion to discuss how the data from this study will be applied to men's clinical decision making. For example, there are examples of DCE questionnaires as part of decision aids for individual patient decision making.
--	--

REVIEWER	Dinkel, Andreas Technische Universität München, Department of Psychosomatic Medicine and Psychotherapy, Klinikum rechts der Isar
REVIEW RETURNED	15-May-2021

GENERAL COMMENTS	This protocol presents the background and the design of a multicenter discrete choice experiment study with patients with metastatic prostate cancer. The authors aim at identifying prostate cancer patients' preferences for treatment options. While such studies using DCE have been conducted previously, research with patients suffering from newly developed metastatic hormone-sensitive prostate cancer is sparse. The is a well-written, clear study protocol that gives sufficient information with regard to the relevance of the topic and the need for such a study. The hypothesis is clearly stated, study design and methods are clearly presented. Patient participation in study development is described, and there is no doubt with regard to ethical standards. I have only a few minor comments:  1. Introduction, p. 10: The authors present very nicely the results of previous studies using discrete choice experiments in prostate cancer. Currently, this is a list of studies and their main results. However, a summary or conclusion is lacking. Did the authors draw any consequences of these examples for study design or methods of their own research? 2. Study hypothesis, p. 10: The authors present the null hypothesis of their study. Clearly, H1 follows from H0, however, the authors might want to state the scientific hypothesis H1 explicitly. 3. Study design, p. 11: Phase 1 will involve interviews with physicians and patients - any predefined characteristics? Do you aim for diversity? With regard to which characteristics? physician's gender, patient's SES,...? 4. Study design, p. 11: Inductive thematic analysis and performance of ITA could be described with a little bit more detail.
---

VERSION 1 – AUTHOR RESPONSE

Reply to Reviewer #1

The authors describe the protocol for a discrete choice experiment (DCE) to determine preferences and values for treatment decisions among men with metastatic, hormone sensitive prostate cancer. The study is clearly described, it is innovative and will move the field of shared decision-making forward. Some minor suggestions for the authors to consider are listed below:

Thank you for your positive review. We have responded in turn to your comments below and made revisions where required in text of the revised manuscript.

Comment #1.1

Page 9 Line 264-283. Qualitative interviews will be used to determine the attributes for the DCE during Phase 1 & 2. It would be helpful to readers to add more detail about the DCE Questionnaire properties (Phase 3). Will there be three levels of options per attribute? How will these levels be determined? Will a specific software be used to create and administer the DCE Questionnaire?

Reply #1.1

Thank you for this pertinent comment. We have now added further details to the Methods and analysis section, on Page 11, line 294. This now reads:

“In the DCE questionnaire, the attributes will be combined into different hypothetical scenarios that will be presented to men in the form of a treatment choice. The hypothetical scenarios included in the questionnaire will be selected using experimental design theory. These attributes and levels will be combined to create treatment scenarios and paired into choice sets of two scenarios. We will use NGENE software (ChoiceMetrics) to generate a 36 choice tasks D-efficient design with non-informative (null) priors and allowing estimation of non-linear effects of attributes [33]. The design is based on the main effects only (i.e. without interactions). The 36 choice tasks will be split into three blocks of 12 tasks to reduce respondent burden. Respondents will be allocated to one of the three blocks. Risk attributes will be presented using icon arrays, ratios, and percentages to ease comprehension (Figure 1) shows an example choice task. The questionnaire will be trialled on a different group of up to 10 men using a semi-structured ‘Think Aloud’ interview and any further changes made to create the final version of the DCE questionnaire (Phase 2) [34]. “

Comment #1.2

Page 11 Line 325. Men will be given the DCE questionnaire to fill in their own time either in the face-to-face or virtual hospital clinic or in their own home. How will patients access to the DCE during clinic encounters or in their own home -- ie will this be a web-based questionnaire? How will the study ensure equal access to patients that do not have access to the internet (will computer access be provided during in person clinic appointments)?

Reply #1.2

We have added clarity regarding how the DCE questionnaire will be completed in the revised manuscript on page 12, line 312. We agree that equity of access is important. For those attending face-to-face clinics tablets will be available to patients to complete the DCE in a similar fashion to those who wish to complete it at home.

This addition now reads:

“The DCE questionnaire will be given to the participants to fill in, either directly following a face-to-face hospital clinic appointment or in their own home. For the electronic version of the survey, REDCap software will be used, with tablets available following a face-to-face hospital clinic appointment where required.”

Comment #1.3

Page 16, Line 474. The trial status is described as follows “IP5-MATTER is planning to open...in January 2021.” This text may benefit from updating prior to publication to reflect the current study status - (i.e. whether the trial opened in January 2021 as expected or whether the start date has been delayed).

Reply #1.3

Thank you for this. We have revised this to read:

“IP5-MATTER is opened to recruitment in 30 centres in England, Wales and Scotland in February 2022 and planned recruitment completed by February 2022 and study completion in April 2022.”

Comment #1.4

Page 15, Line 461-465. The results of this study certainly have implications for healthcare providers, clinicians and patients. The authors can consider expanding the discussion to discuss how the data from this study will be applied to men’s clinical decision making. For example, there are examples of DCE questionnaires as part of decision aids for individual patient decision making.

Reply #1.4

Thank-you for highlighting DATs. We agree there is ongoing exciting research into the role of DCE in DAT development. We have now edited the text in the Discussion, on Page 17, and the last paragraph reads as follows:

“Furthermore, it is possible that once such treatment pathways are established the findings from this DCE may be integrated into further work towards the creation of a decision treatment aid (DAT) for men with mHSPC. There is residual uncertainty in the role of DCE’s in the development of such DATs [36]. However, this methodology is currently being explored in parallel to this study in localised prostate cancer and other benign surgical settings, and if proven IP5-MATTER may offer utility in the development in any future decision treatment aid for this cohort [37,38].”

Reply to Reviewer 2

This protocol presents the background and the design of a multicenter discrete choice experiment study with patients with metastatic prostate cancer. The authors aim at identifying prostate cancer patients’ preferences for treatment options. While such studies using DCE have been conducted previously, research with patients suffering from newly developed metastatic hormone-sensitive prostate cancer is sparse.

The is a well-written, clear study protocol that gives sufficient information with regard to the relevance of the topic and the need for such a study. The hypothesis is clearly stated, study design and methods are clearly presented. Patient participation in study development is described, and there is no doubt with regard to ethical standards.

I have only a few minor comments:

We thank reviewer #2 for his positive review of our manuscript. We have responded to his comments in turn below and made alterations in text of the revised manuscript where required.

Comment #2.1

Introduction, p. 10: The authors present very nicely the results of previous studies using discrete choice experiments in prostate cancer. Currently, this is a list of studies and their main results.

However, a summary or conclusion is lacking. Did the authors draw any consequences of these examples for study design or methods of their own research?

Reply #2.1

We thank the reviewer for drawing our attention to this; we have now revised and added the following paragraph in the Previous discrete choice experiments in prostate cancer section, on Page 8, and the last paragraph reads as follows:

“In summary, to our knowledge, no studies have undertaken a DCE to elicit men’s preferences for systemic therapy in combination with either local cytoreductive and/or metastasis-directed treatments. The aforementioned, DCE studies in prostate cancer (non-metastatic, metastatic castrate-resistant, and metastatic hormone-sensitive prostate cancer) have predominantly highlighted patients’ preferences in the context of being offered established systemic therapy options and a select number of bone-targeted agents.

Therefore, the overall understanding of how patients value novel treatment options in this setting does remain limited. The IP5-MATTER study is a multicentre DCE designed to evaluate novel treatments (cytoreductive prostatectomy, external beam radiotherapy, minimally-invasive ablative therapy and SABR) in addition to systemic therapy for the first time. To appropriately invest in emerging cytoreductive prostate and metastasis-directed treatments options that are most acceptable to such patients, we believe that an attempt to formalise our understanding of the trade-offs between oncological benefits and risks in this cohort of patients should be performed.”

Comment #2.2

Study hypothesis, p. 10: The authors present the null hypothesis of their study. Clearly, H1 follows from H0, however, the authors might want to state the scientific hypothesis H1 explicitly.

Reply #2.2

We have stated the scientific hypothesis clearly for the readership in the revised manuscript. This now reads:

“The scientific hypothesis (H1) is that men with de novo synchronous hormone-sensitive metastatic prostate cancer may not accept the complications and risk of side effects associated with cytoreductive local prostate and/or metastasis directed therapy for any potential oncological benefit.”

Comment #2.3

Study design, p. 11: Phase 1 will involve interviews with physicians and patients - any predefined characteristics? Do you aim for diversity? With regard to which characteristics? physician’s gender, patient’s SES,...?

Reply #2.3

Thank you for raising this point. In Phase 1 of the study the interviews with physicians will be drawn from the multidisciplinary team meeting for prostate cancer. This is a diverse pool of physicians and specialist nurses. We do not have any other predefined characteristics. Patients involved in phase I and II of the study will be drawn from patients who have had a diagnosis of metastatic hormone sensitive prostate cancer at our tertiary centre.

We have included this information on Page 14 of the revised text, this also appears below:

“The initial qualitative phase (Phase 1) will involve semi-structured interviews with (up to 5) clinician member of the multidisciplinary or tumour board meeting and (up to 5) patients who have had a diagnosis of mHSPC, to identify and define the key attributes associated with treatment options that would warrant trade-off evaluation. These will be used to create the first version of the questionnaire (Phase 1) [25].”

For the main phase 3 DCE survey, we will collect patient characteristics understand the diversity of respondents and explore the heterogeneity of preferences. The protocol did not list all characteristics that we planned to collect data on. We will collect information on age and education and many others that we think could explain the heterogeneity.

We have included this information on Page 14 of the revised text, this also appears below:

“The final survey instrument will also collect information on patients’ characteristics (age, education, etc.)”

Comment #2.4

Study design, p. 11: Inductive thematic analysis and performance of ITA could be described with a little bit more detail.

Reply #2.4

Thank you for highlighting this, we have added further detail to the section on ITA with a new citation. This appears on page X, line x of the revised manuscript and also below:

“Inductive thematic analysis will be performed as described by Nowell and colleagues 6-step iterative process [REF]. Researchers will i) Familiarize themselves with the data (e.g. document potential themes/codes) ii) Generate initial codes (e.g. reflective journaling) iii) Search for themes (e.g. triangulation of data) iv) Review themes (e.g. test for referential adequacy to raw data) v) Define themes and name, and vi) Produce final report [REF]”.

Nowell, Lorelli S., Jill M. Norris, Deborah E. White, and Nancy J. Moules. "Thematic analysis: Striving to meet the trustworthiness criteria." *International journal of qualitative methods* 16, no. 1 (2017): 1609406917733847.

Additional References included in Revised Manuscript

Dowsey, M. M., Scott, A., Nelson, E. A., Li, J., Sundararajan, V., Nikpour, M., & Choong, P. F. M. (2016). Using discrete choice experiments as a decision aid in total knee arthroplasty: Study protocol for a randomised controlled trial. *Trials*, 17(1).

Genie, M. G., Nicoló, A., & Pasini, G. (2020). The role of heterogeneity of patients’ preferences in kidney transplantation. *Journal of Health Economics*, 72, 102331.

Rose, J. M., & Bliemer, M. C. J. (2013). Sample size requirements for stated choice experiments. *Transportation*, 40(5), 1021–1041.

Nowell, Lorelli S., Jill M. Norris, Deborah E. White, and Nancy J. Moules. "Thematic analysis: Striving to meet the trustworthiness criteria." *International journal of qualitative methods* 16, no. 1 (2017): 1609406917733847.

National Cancer Research Institute, (NCRI). NCRI Prostate Group Annual Report 2019-20. [Putting men’s preferences at the center of the doctor-patient relationship: The Prostate cAncER Treatment prEfeRences (PARTNER) Test]. 2020. Available at: <https://www.ncri.org.uk/wp-content/uploads/NCRI-Prostate-Group-Annual-Report-2019-20.pdf>

VERSION 2 – REVIEW

REVIEWER	Lane, Giulia University of Michigan
REVIEW RETURNED	09-Sep-2021
GENERAL COMMENTS	The authors have answered all my previous comments satisfactorily. I have no further comments